# Evaluation of Au/ZrO₂ Catalysts Prepared via Postsynthesis Methods in CO₂ Hydrogenation to Methanol

**Tatiparthi Vikram Sagar [1], Janez Zavašnik [2], Matjaž Finšgar [3], Nataša Novak Tušar [1] and Albin Pintar [1,*]**

[1] Laboratory for Environmental Sciences and Engineering, Department of Inorganic Chemistry and Technology, National Institute of Chemistry, Hajdrihova 19, SI-1001 Ljubljana, Slovenia; tvikramsagar@gmail.com (T.V.S.); natasa.novak.tusar@ki.si (N.N.T.)

[2] Department of Gaseous Electronics, Jožef Stefan Institute, Jamova 39, SI-1000 Ljubljana, Slovenia; janez.zavasnik@ijs.si

[3] Faculty of Chemistry and Chemical Engineering, University of Maribor, SI-2000 Maribor, Slovenia; matjaz.finsgar@um.si

\* Correspondence: albin.pintar@ki.si; Tel.: +386-1476-0237; Fax: +386-1476-0460

**Abstract:** Au nanoparticles supported on ZrO₂ enhance its surface acidic/basic properties to produce a high yield of methanol via the hydrogenation of CO₂. Amorphous ZrO₂-supported 0.5–1 wt.% Au catalysts were synthesized by two methods, namely deposition precipitation (DP) and impregnation (IMP), characterized by a variety of techniques, and evaluated in the process of CO₂ hydrogenation to methanol. The DP-method catalysts were highly advantageous over the IMP-method catalyst. The DP method delivered samples with a large surface area, along with the control of the Au particle size. The strength and number of acidic and basic sites was enhanced on the catalyst surface. These surface changes attributed to the DP method greatly improved the catalytic activity when compared to the IMP method. The variations in the surface sites due to different preparation methods exhibited a huge impact on the formation of important intermediates (formate, dioxymethylene and methoxy) and their rapid hydrogenation to methanol via the formate route, as revealed by means of in situ DRIFTS (diffuse reflectance infrared Fourier transform spectroscopy) analysis. Finally, the rate of formation of methanol was enhanced by the increased synergy between the metal and the support.

**Keywords:** Au/ZrO₂; CO₂ hydrogenation; methanol synthesis; in situ DRIFTS; CO₂ utilization

## 1. Introduction

The production of carbon-based pollutants through the utilization of fossil fuels has increased enormously since the industrial revolution. Strategies to reduce CO₂ emissions in the atmosphere are essential [1–4]. In this regard, CO₂ is considered to be an abundant C1 source [5] that can be used as a raw material to produce vital value-added products, such as syngas, dimethyl ether, formic acid, methane, higher hydrocarbons, and methanol, etc. [6–8]. Among all of the CO₂ conversion routes, most studies in recent years have been focused on CO₂ hydrogenation to methanol [9]. Methanol has several important applications, including usage in the plastic industry and as a solvent in many important organic reactions [1,10]. The main use of methanol is as a primary feedstock for the production of different commodity chemicals [11]. Currently, industrial methanol production is carried out using synthesis gas (CO/CO₂/H₂) with Cu/ZnO/Al₂O₃-based catalysts at high pressure (50−100 bar) [8,12–14]. The CO and CO₂ hydrogenation reactions require high pressure to ensure the maximum conversion and high selectivity towards methanol; hence, the research focuses on the development of efficient catalysts for direct CO₂ hydrogenation to methanol. The important reactions of methanol production are as follows:

$$CO_2 + 3H_2 \rightarrow CH_3OH + H_2O \rightarrow \Delta H° = -49.5 \text{ kJ mol}^{-1} \qquad (1)$$

$$CO_2 + H_2 \rightarrow CO + H_2O \rightarrow \Delta H° = +41.2 \text{ kJ mol}^{-1} \qquad (2)$$

The catalysts for the hydrogenation of $CO_2$ to methanol include, among others, supported Au particles, $In_2O_3$, Ni-Ga, Pd-Ga, Zn-Zr and Mn-Co [15,16]. Due to its availability, copper has been widely investigated for methanol production from $CO_2$. Numerous copper catalysts have been designed, and successful examples are $Cu/ZnO/Al_2O_3$ (used industrially for the hydrogenation of CO and $CO_2$), $Cu/ZnO/ZrO_2$, $Cu/ZnO/TiO_2$, $Cu/ZrO_2$ and $Cu/CeO_2$ [13,17–21]. In addition, supported copper usually suffers from deactivation caused by nanoparticle sintering under harsh reaction conditions [22,23].

Recently, catalysts developed using Au as an active metal have been proposed, which are related to the enhanced electron transfer from the support to the metal. The negatively charged gold nanoclusters supported on a suitable metal oxide enhance the activation of $CO_2$ molecules [24]. Besides this, other studies related to Au-supported catalytic systems showed high selectivity towards methanol compared to Cu-based catalytic formulations [24–27]. Changes associated with the electronic properties of gold modify the activity by influencing the adsorption of reactants and the activation of intermediates, and $Au^{\delta+}$ promotes the dissociation of $H_2$ molecules [27,28]. According to Hartadi et al. [29], the particle size of the active metal has a huge impact on the catalytic performance. They further reported that the existence of Au nanoparticles leads to an Au-rich surface that improves the selectivity to methanol [29] and efficiency at lower temperatures (<170 °C) [24]. Different studies related to the Au-based catalysts have been carried out recently [30–39]. Theoretical studies on Au supported on carbide supports such as TiC and MoC show a remarkable activity of $CO_2$ reduction to methanol with Au due to the enhancement of the ability of noble metals to bond and activate $CO_2$ [31,33]. An experimental study on Au supported on $CeO_2$ and ZnO showed the importance of the support and metal–support interactions for the reaction of methanol production [37]. In this study, it was demonstrated that the over-reduction of the support could result in the deactivation of the catalyst. Another study based on different supported oxides with Au shows the importance of the selection of a support to promote the reaction [38]. For instance, $TiO_2$ and $Fe_2O_3$ supports exclusively produce CO and methane. On the other hand, ZnO and $CeO_2$ enable the high production of methanol. A strong metal–support interaction between Au and $In_2O_3$ leads to an important reactive $Au^{\delta+}–In_2O_{3-x}$ interface for the activation and hydrogenation of $CO_2$ to methanol [27]. A study on a series of Au-CuO/SBA-15 catalysts showed the importance of Au nanoparticles, which exhibited higher catalytic activity compared to CuO/SBA-15 [32].

Mechanistic studies in the presence of Au-based catalysts were conducted using different techniques, such as in situ DRIFTS (diffuse reflectance infrared Fourier transform spectroscopy) and near-ambient-pressure X-ray photoelectron spectroscopy (NAP-XPS), as well as DFT analyses; it was reported that the $CO_2$ hydrogenation to methanol reaction occurs through the formation of surface intermediates, namely formate and methoxy species [9,39–42]. The formation of a formate intermediate occurs rapidly over reducible supports [41], but further hydrogenation steps depend on the above-stated factors, such as the size of active metal ensembles and their acidic/basic properties. However, the acidic/basic properties and their role in improving the conversion of $CO_2$ to methanol has to be studied further in order to develop a highly active catalyst.

In this study, the authors mainly focus on the surface changes that occur due to the synthesis of Au-based $ZrO_2$ catalysts with two different preparation methods: deposition precipitation (DP) and impregnation (IMP). The influence of the surface changes observed with the method of preparation was investigated by using different physico-chemical techniques, and the advantages of these surface modifications to the methanol production by $CO_2$ hydrogenation were mainly the focus. This work also presents differences in the reaction mechanism attributed to different methods of catalyst preparation through in situ DRIFTS analysis.

## 2. Results and Discussion

### 2.1. Catalytic Activity

The results of the $CO_2$ hydrogenation reaction to methanol examined in a conventional high-pressure fixed-bed reactor in the presence of $ZrO_2$-supported Au catalysts prepared by the deposition precipitation and impregnation methods are depicted in Figure 1.

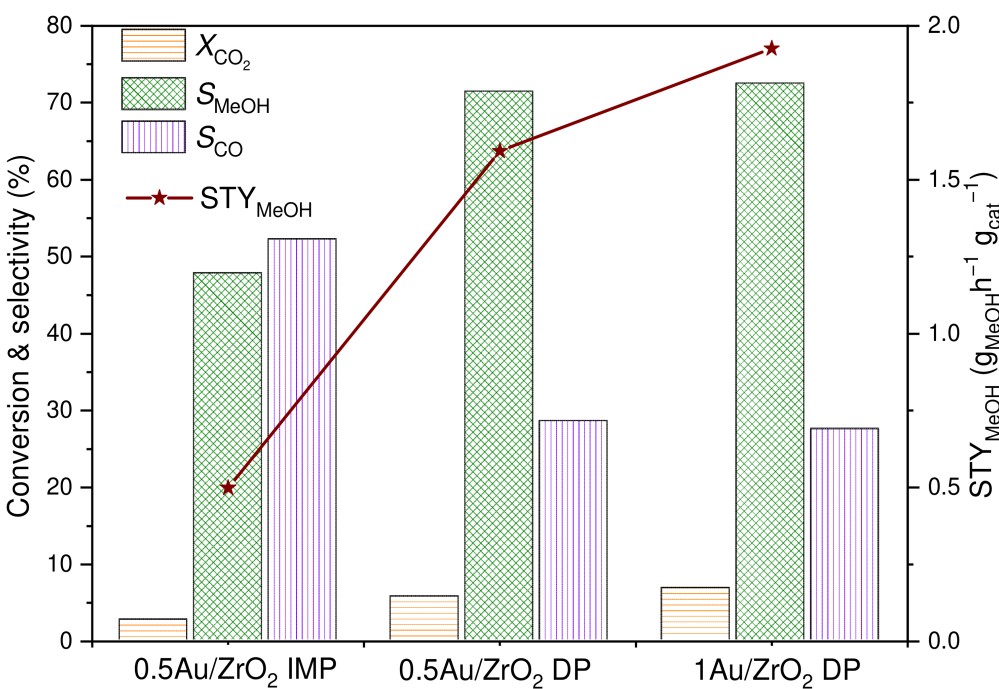

**Figure 1.** Comparison of the catalytic activity of $Au/ZrO_2$ catalysts synthesized by different preparation procedures. Operating conditions: T = 240 °C; $P_{tot.}$ = 40 bar; catalyst weight: 50 mg (homogeneously mixed with 200 mg of SiC); gas flow rate: 100 mL/min; composition of the feed gas stream: 24 vol.% $CO_2$, 72 vol.% $H_2$ (balanced with $N_2$).

Both catalysts synthesized by the DP method showed a high conversion and selectivity of methanol compared to the catalyst synthesized by the IMP method. For DP-method catalysts, the increase in Au content increased the conversion of $CO_2$ from 5.7 to 6.8%. However, the resulting selectivity of methanol appeared to be similar, with values of 71.5 and 72.5% for the catalyst samples with 1 and 0.5 wt.% Au loading, respectively. The STY of methanol increased from 1.6 to 1.9 $g_{MeOH}$ $h^{-1}$ $g_{cat}^{-1}$ with the increase of the Au content. On the other hand, the 0.5 wt.% $Au/ZrO_2$ catalyst synthesized by the IMP method showed the lowest activity (2.7% $CO_2$ conversion) and less than 50% selectivity towards methanol. However, the highest selectivity towards the formation of CO as a side product was observed in the presence of this catalyst sample (Figure 1). The results obtained in this study are promising and in good agreement with the performance of Au-based catalysts reported in the literature (Table S1, Supplementary Information). The high WHSV value used in the performed experiments (i.e., 120,000 $cm^3$ $h^{-1}$ $g_{cat}^{-1}$) resulted in a high space–time yield. The carbon balance of the evaluated catalysts is 98, 96 and 96% for 0.5 $Au/ZrO_2$ IMP, 0.5 $Au/ZrO_2$ DP and 1 $Au/ZrO_2$ DP, respectively.

Figure S1a (Supplementary Information) shows the activity data of the 0.5 $Au/ZrO_2$ DP catalyst measured at 50 bar and different temperatures (from 200–260 °C) with an increment of 20 °C. With the increase of temperature there was an increase in $CO_2$ conversion, while the methanol selectivity decreased continuously with the temperature rise. On the other hand, the STY yield increased up to 240 °C and decreased during the further increase of the temperature. As such, we performed a time-on-stream test to examine the catalyst stability at this particular temperature. It can be seen in Figure S1b that the

0.5 Au/ZrO$_2$ DP catalyst exhibited constant performance up to 63 h of stability study. The CO formed during the reaction (Figure S1a) may lead to poising of gold clusters on the catalyst surface, and may influence hydrogen adsorption, resulting in low conversion of CO$_2$. However, this seems to be of less significance because no catalyst deactivation was observed during the stability test (Figure S1b).

### 2.2. Catalyst Characterization

2.2.1. N$_2$ Physisorption, Au Loading

Figure 2 shows the N$_2$ adsorption–desorption isotherms of reduced catalysts. All of these isotherms resemble a type-I isotherm that represents microporous materials. An H1-type hysteresis loop often signifies that the porous materials consist of well-defined cylindrical-like pore channels and/or agglomerates of approximately uniform spheres [43]. The catalysts exhibited a high specific surface area, total pore volume and average pore diameter (Table 1). The catalysts prepared by the DP method show similar values of specific surface area (SSA), total pore volume and average pore diameter, regardless of the Au loading. However, these catalysts show a small decrease in SSA (of about 11%) in comparison to the bare ZrO$_2$ support, while the IMP-prepared sample shows an SSA decrease of about 35%, which we attribute to the blockage of pores by Au crystallites. A small increase of the average pore diameter (Table 1) may be attributed to the blockage of narrow pores in the ZrO$_2$ support by deposited Au ensembles.

The values listed in Table 1 demonstrate the good agreement between the nominal (0.5 or 1.0 wt.%) and actual Au loading in the synthesized catalyst samples.

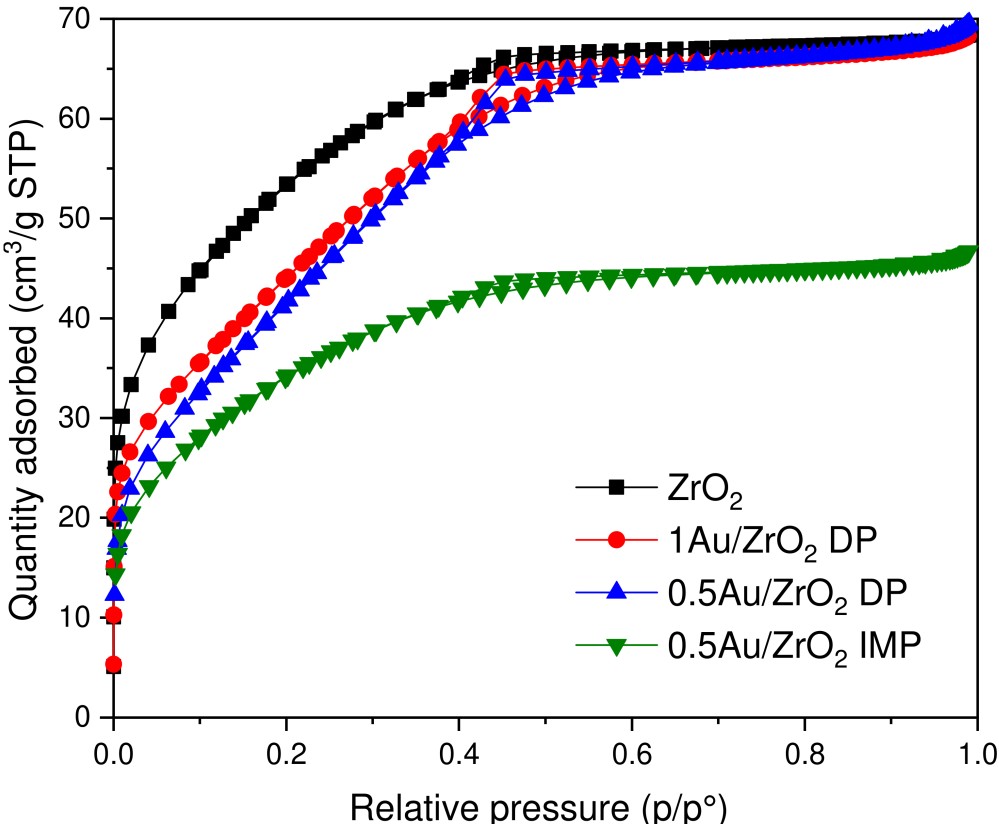

**Figure 2.** N$_2$ adsorption–desorption isotherms of ZrO$_2$ support and ZrO$_2$-supported Au deposition precipitation/impregnation catalysts after reduction.

**Table 1.** Specific surface area (SSA), total pore volume, average pore diameter and actual Au content of the ZrO$_2$ support and Au deposition precipitation/impregnation catalysts after reduction.

| Catalyst | SSA (m$^2$/g) | Total Pore Volume (cm$^3$/g) | Average Pore Diameter (Å) | Au Content (wt.%) |
|---|---|---|---|---|
| ZrO$_2$ | 187 | 0.115 | 22.5 | - |
| 1 Au/ZrO$_2$ DP | 166 | 0.105 | 25.2 | 0.96 |
| 0.5 Au/ZrO$_2$ DP | 168 | 0.110 | 26.2 | 0.49 |
| 0.5 Au/ZrO$_2$ IMP | 122 | 0.072 | 23.4 | 0.52 |

2.2.2. X-ray Powder Diffraction Analysis

The phase composition of the catalysts reduced at 350 °C was analyzed by X-ray powder diffraction (Figure 3). In the XRD diffractogram of the ZrO$_2$ support, we observed a broad peak at 2θ ≅ 30° and a second, lower-intensity peak at 2θ ≅ 50°; these diffractions were ascribed to the amorphous ZrO$_2$ [44–48]. The addition of Au ensembles to the amorphous ZrO$_2$ support showed sharp peaks at 2θ = 30.3, 35.3, 50.3 and 60.1°; these peaks were attributed to the crystallization of the amorphous ZrO$_2$ support, i.e., the phase transformation of the amorphous to the tetragonal phase of ZrO$_2$ (JCPDS card 01-080-0784). This phase transformation was perhaps induced due to the enhanced interaction between the metal and the support. A close observation of the intensities of these peaks reveals that they vary with the catalyst preparation method. Here, the catalysts synthesized by the DP method show high-intensity peaks compared to the solid synthesized by the IMP method. This indicates that the metal–support interactions are enhanced with the DP preparation method. It should be further noted that the Au cubic (fcc) phase was observed at 2θ = 38.2, 44.4. 64.5, 77.6 and 81.8° (JCPDS Card No. 04-0784) only in the IMP synthesized catalyst but not in the case Au/ZrO$_2$ DP samples. The identification of the Au cubic phase in the 0.5 Au/ZrO$_2$ IMP catalyst indicates the bulk cluster formation of Au in the case of the IMP method of preparation, in spite of the low Au loading. This is in accordance with the results of the N$_2$ physisorption analysis (Table 1 and Figure 2), in which we observed a severe pore blockage in the 0.5 Au/ZrO$_2$ IMP catalyst.

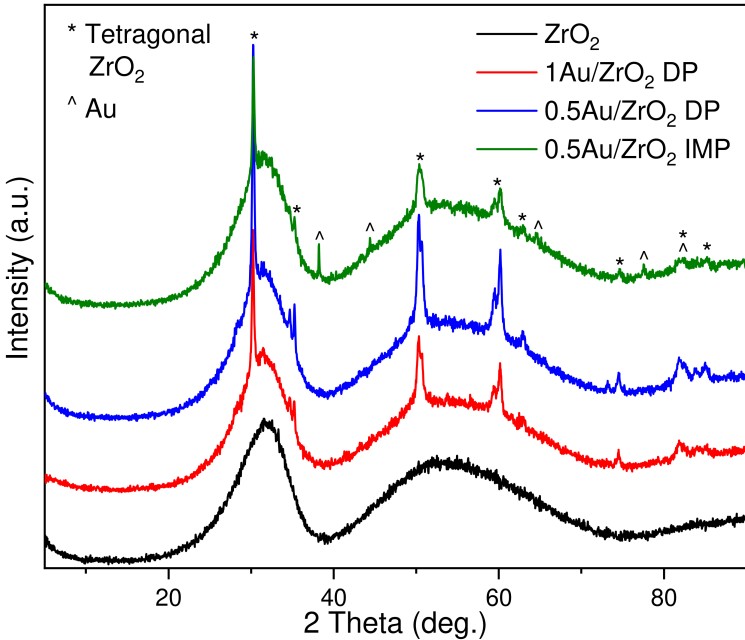

**Figure 3.** X-ray powder diffraction patterns of the ZrO$_2$ support and Au/ZrO$_2$ deposition precipitation/impregnation catalysts after reduction.

The in situ X-ray diffraction study of the reduction of amorphous $ZrO_2$ was conducted with 5% $H_2$-balanced Ar at temperatures ranging from 25 to 500 °C; the obtained results are depicted in Figure 4. From the inset graph of Figure 4, we can conclude that the amorphous $ZrO_2$ shows no phase transformation during the temperature rise from 25 to 350 °C. The first phase transformation was observed at T = 400 °C, and continued to high crystallinity with the further temperature increase, during which we observed high-intensity patterns at 2θ = 30.5, 35.3, 50.7, 60.2 and 74.5°. These XRD patterns were related to the JCPDS file 01-080-0784 of the tetragonal $ZrO_2$ phase. Based on the above, we can conclude that the phase transformation of amorphous $ZrO_2$ to the tetragonal $ZrO_2$ phase occurring at 350 °C is mainly due to the interaction with Au metallic ensembles deposited on the catalyst surface. This is in agreement with the study of Horti et al., who investigated $ZrO_2$ crystallization as a function of temperature, and observed that the crystallization of amorphous $ZrO_2$ to different crystallographic forms of $ZrO_2$ is dependent on the temperature of the calcination [46].

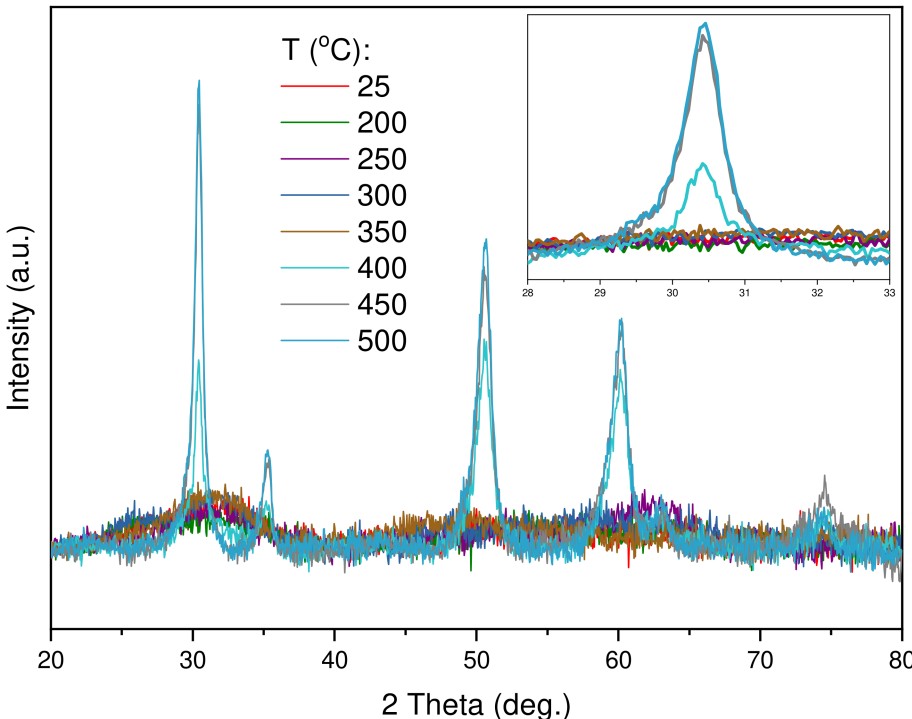

**Figure 4.** X-ray powder diffraction patterns obtained during the in situ reduction of the $ZrO_2$ support in the temperature range of 25–500 °C.

### 2.2.3. Pyridine Temperature-Programmed Desorption (Py-TPD)

A classical technique of the temperature-programmed desorption of a basic probe molecule was used in the present study in order to obtain a quantitative estimation of the total number and nature of acidic sites on the surface of the reduced catalysts. Usually, a suitable basic probe is pre-adsorbed on the catalyst surface until dynamic saturation, followed by desorption with an increasing temperature. In this study, pyridine was used as a basic probe molecule, and its adsorption and desorption were monitored by a TG device (the change of weight). Furthermore, the obtained derivatives of weight are presented in Figure 5, while Table 2 shows the quantified total acidity values of the reduced catalysts. The bare $ZrO_2$ support showed high total acidity with 0.2571 μmol/g uptake of pyridine compared to Au-containing catalysts. The values of the total acidity of 0.1737, 0.1528 and 0.1221 μmol/g were obtained for 0.5 Au/$ZrO_2$ IMP, 0.5 Au/$ZrO_2$ DP and 1 Au/$ZrO_2$ DP catalyst samples, respectively. However, it is clear from Figure 5 that the strength of the surface acidic sites is different for each catalyst. Concerning the bare $ZrO_2$ support, different desorption peaks were observed at 248, 402, 458 and 481 °C. The first low-temperature

desorption peak is due to the early desorption of pyridine from the surface; this little hump is due to the presence of weak acidic sites. Furthermore, the sharp high-intensity and a low-intensity desorption peaks occurring at 402 and 458 °C are due to the moderately strong acidic sites [49–53], and finally, the peak at 481 °C is due to the phase change from the amorphous to the cubic $ZrO_2$ phase. On the other hand, the addition of 0.5 wt.% Au showed a noticeable change in the character of the acidic sites. In both 0.5 Au/$ZrO_2$ DP and 0.5 Au/$ZrO_2$ IMP catalysts, three main desorption peaks were observed at 243, 473 and 610/624 °C (IMP/DP $T_{max}$). The weak acidic sites were observed as it was in the case of the bare $ZrO_2$ support, but the moderately strong acidic sites were shifted to higher temperatures, indicating that the strength of the acidic sites was increased with the Au addition, which is due to the change in the metal–support interaction. A new desorption peak appeared at 610 and 624 °C for IMP and DP-based 0.5 Au/$ZrO_2$ catalysts, respectively. This high-temperature peak indicates the presences of strong acidic sites. It can be seen that the 0.5 Au/$ZrO_2$ DP catalyst exhibits stronger acidic sites compared to the 0.5 Au/$ZrO_2$ IMP sample. Finally, Figure 5 demonstrates that the 1 Au/$ZrO_2$ DP catalyst shows only one pyridine desorption peak, which appears at a high temperature, i.e., 618 °C. This implies that there were only strong acidic sites present on the surface of this solid.

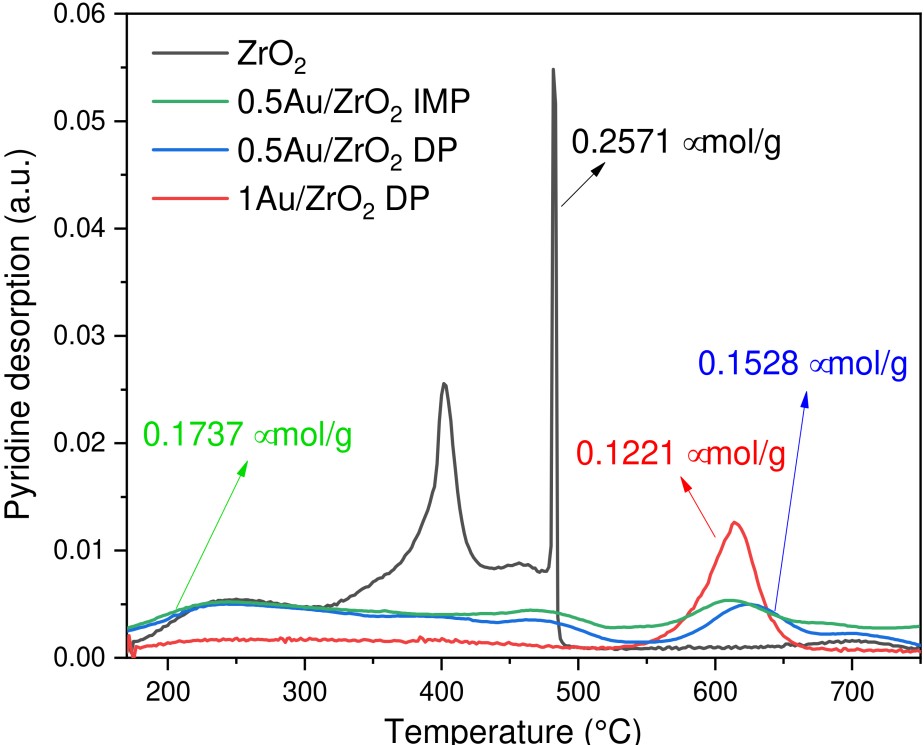

**Figure 5.** Py–TPD curves of the $ZrO_2$ support and Au/$ZrO_2$ deposition precipitation/impregnation catalysts after reduction.

**Table 2.** Total acidity and total basicity of the $ZrO_2$ support and catalyst samples, as determined by means of the temperature-programmed desorption of pyridine and $CO_2$.

| Catalyst | Total Acidity (μmol/g) | Total Basicity (μmol/g) | γ Peak Area (a.u.) |
|---|---|---|---|
| $ZrO_2$ | 0.2571 | 47.02 | 33.69 |
| 1 Au/$ZrO_2$ DP | 0.1528 | 62.67 | 45.29 |
| 0.5 Au/$ZrO_2$ DP | 0.1221 | 63.96 | 43.39 |
| 0.5 Au/$ZrO_2$ IMP | 0.1737 | 38.12 | 29.80 |

### 2.2.4. CO$_2$ Temperature-Programmed Desorption (CO$_2$-TPD)

The TPD of CO$_2$ on reduced Au/ZrO$_2$ catalysts was carried out by the low-temperature adsorption of an acidic probe molecule (CO$_2$), which selectively binds to the basic sites on the catalyst surface and then desorbs with an increasing temperature. These measurements help us to understand the nature of basic sites and the total basicity of the examined catalyst samples [7].

The obtained results of CO$_2$-TPD measurements performed in the temperature range of 50 to 350 °C are depicted in Figure 6. It can be seen that the great majority of CO$_2$ was desorbed at temperatures up to 300 °C. This might be the reason why, during catalytic runs, low CO$_2$ conversions were measured at 240 °C and above (Figure S1a). A broad peak (Figure 6) with a long tail ranging from 50 to 325 °C was deconvoluted into three Gaussian peaks designated as α, β and γ peaks. According to the previous research reports, the different (low, medium and high) strengths of basic sites are ascribed to the presence of hydroxyl groups, cus O$^{2-}$ centres, and acid–base pairs (i.e., Zr$^{4+}$/O$^{2-}$ centres) [54,55]. In the present study, the high strength of the basic sites for CO$_2$ adsorption plays an important role as an active site for methanol synthesis in the process of CO$_2$ hydrogenation compared to the low- and medium-strength of the basic sites. This is because high-strength basic sites desorb CO$_2$ at temperatures similar to the operating temperature window in which the reaction was performed (i.e., 200–260 °C). The maximum temperature position of the γ peak was observed from 175 to 185 °C for all of the catalysts. The highest T$_{max}$ of 185 °C was found in the case of the ZrO$_2$ support; the addition of Au exhibited a tendency of decreasing T$_{max}$ values to 175–180 °C. This observation evidences the ease of CO$_2$ desorption and the further participation in the hydrogenation reaction, which is enhanced in the presence of Au/ZrO$_2$ solids.

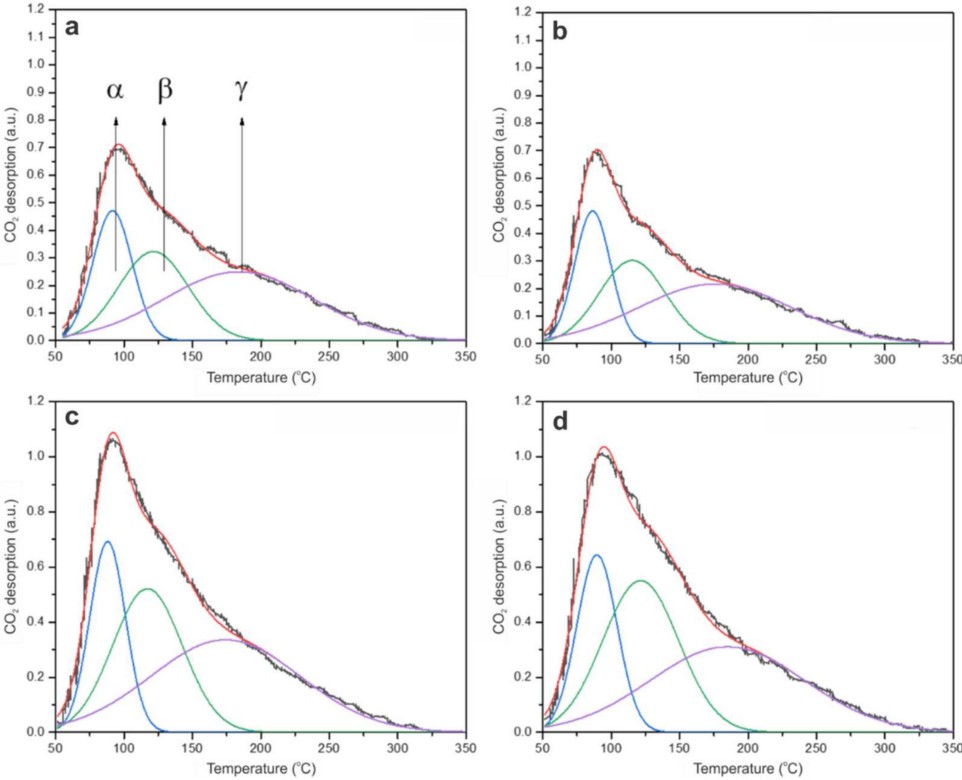

**Figure 6.** CO$_2$–TPD curves of the ZrO$_2$ support and Au/ZrO$_2$ deposition precipitation/impregnation catalysts after reduction: (**a**) ZrO$_2$, (**b**) 0.5 Au/ZrO$_2$ IMP, (**c**) 1 Au/ZrO$_2$ DP, (**d**) 0.5 Au/ZrO$_2$ DP.

The values of the total basicity are listed in Table 2. The bare ZrO$_2$ support showed a value of 47.02 μmol/g total basicity. In DP-based Au/ZrO$_2$ catalysts, the total basicity is increased to the values of 62.67 and 63.96 μmol/g, with an increase of Au content to 0.5

and 1 wt.%, respectively. On the other hand, the catalyst prepared by the impregnation method (0.5 Au/ZrO$_2$ IMP) showed a decrease in total basicity to a value of 38.12 µmol/g.

The areas of the deconvoluted $\gamma$ peak from Figure 6 are reported in Table 2. It can be observed that the number of basic sites corresponding to acid–base pairs (i.e., Zr$^{4+}$/O$^{2-}$ centres) increased in the Au/ZrO$_2$ DP catalysts with the increase of Au content to 43 to 45 a.u. for 0.5 and 1 wt.% Au loadings, respectively. On the other hand, in the case of the 0.5 Au/ZrO$_2$ IMP catalyst, the area of the deconvoluted $\gamma$ peak is equal to 30 a.u.; this indicates a decrease in the number of acid–base pair sites with the utilization of the impregnation method for the deposition of Au ensembles on the catalyst surface that might be due to a lesser extent of metal–support interactions.

### 2.2.5. UV-Vis Diffuse Reflectance Spectroscopy

The UV-Vis diffuse reflectance (DR) spectra of synthesized samples are depicted in Figure S2 (Supplementary Information). This technique was essential in order to understand the metal environment and the ligand-to-metal charge transfer at an electronic level. The ZrO$_2$ support shows a sharp peak at 252 nm, which was attributed to the charge transfer of O$^{2-}$ to Zr$^{4+}$ bulk zirconia [56,57]. Two absorption bands were further observed in the Au-based catalysts. These bands are located in the region of 225–250 nm that refers to the charge transfer from a ligand to the metal in the support. Along with this, another absorption band was also observed at around 560 nm, which was attributed to the surface plasmon absorption of spatially confined electrons in Au$^0$ metallic gold nanoparticles [58,59]. From Figure S2, we can conclude that all of the Au-containing catalysts synthesized with different preparation methods show similar peaks, which confirms the presence of metallic Au nanoparticles.

### 2.2.6. TEM Analysis

The results of the TEM analysis of the Au/ZrO$_2$-reduced catalysts are summarized in Figure 7. The ZrO$_2$ matrix is amorphous, and the Au NPs are distributed on the ZrO$_2$ surface. Au NPs are single crystals with a facetted morphology, and show distinct internal defects. The size of the Au NPs, as measured from the TEM micrographs, is summarized in Table 3, and strongly depends on the synthesis method. Au-based catalysts synthesized by the DP method show a lower mean Au particle size compared to the solid prepared by the IMP method (Figure S3, Supplementary Information). The 0.5 Au/ZrO$_2$ DP catalyst exhibits an Au particle size distribution from 0.5 to 2.0 nm, with a mean particle size of 1.1 nm. Similarly, the 1 Au/ZrO$_2$ DP catalyst shows an Au particle size distribution from 1 to 30 nm; however, the mean particle size increases with the increase of Au loading, and equals 10.8 nm. On the other hand, the IMP method synthesized catalyst with 0.5 wt.% Au loading shows bulk clusters of Au ranging from 10 to 120 nm, with a mean particle size of 50.1 nm. This observation is in very good agreement with the results of both (i) XRD examination (Figure 3), where we observed the formation of the Au cubic phase only in the case of 0.5 Au/ZrO$_2$ IMP catalyst, and (ii) N$_2$ physisorption, where the same catalyst showed the lowest specific surface area (Table 1) due to the high extent of pore blockage. Based on the above observations, one can conclude that the bulk formation of Au nanoparticles on the surface of the ZrO$_2$ support was observed with the impregnation method.

Figure 1 shows that the 1 Au/ZrO$_2$ DP sample exhibits higher activity compared to the 0.5 Au/ZrO$_2$ DP solid. We relate the high activity of the Au/ZrO$_2$ catalyst with 10.8 nm particle size (i.e., 1 Au/ZrO$_2$ DP sample) to a very high surface-to-volume (SA:V) ratio, which exponentially grows for the NPs < 10 nm in size. At about 10 nm, the Au NPs have already partially developed their morphology, which additionally contributes to increased SA:V [34]. Additionally, the Au loading for sample 1 Au/ZrO$_2$ DP was 2× higher as for 0.5 Au/ZrO$_2$ DP sample. The high activity of the referred sample is therefore a combination of the "appropriate" size of the NPs, their morphology, and higher loading, as for other samples.

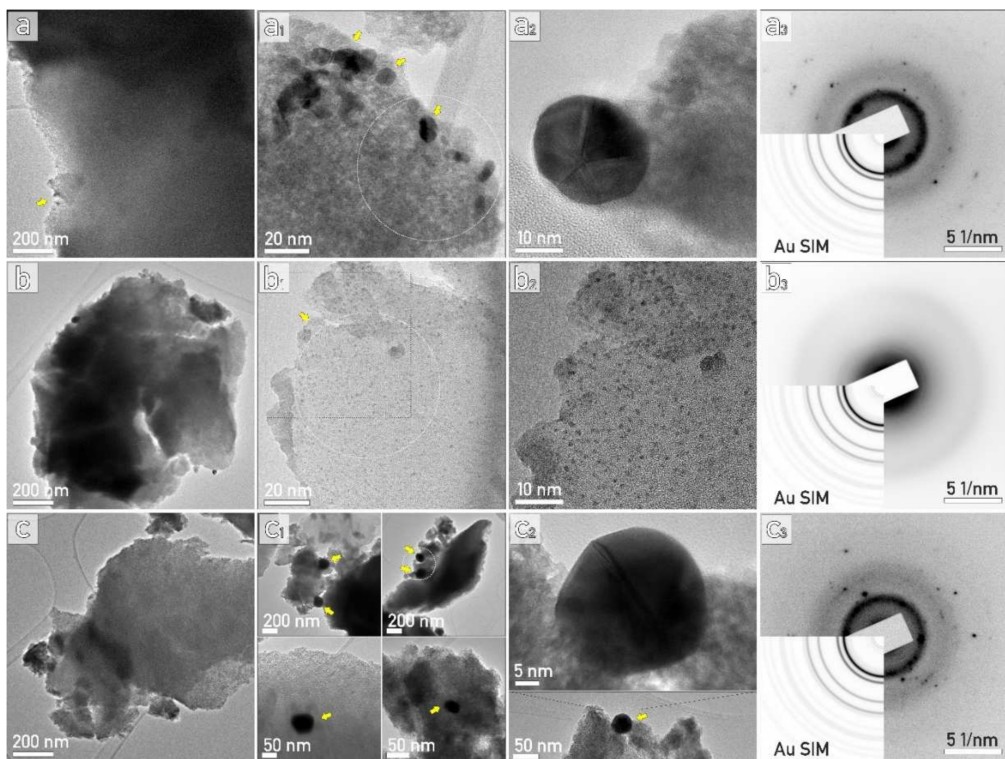

**Figure 7.** Overview TEM micrographs of (**a**) 1 Au/ZrO$_2$ DP, (**b**) 0.5 Au/ZrO$_2$ DP, and (**c**) 0.5 Au/ZrO$_2$ IMP catalyst samples. The second column (**a1**,**b1**,**c1**) shows the identified Au NPs (marked by arrows). The white circle marks the approximate size of the region where the diffraction patterns were recorded. (**a2**,**b2**,**c2**) Au NPs. (**a3**,**b3**,**c3**) Selected-area electron diffraction pattern, with the simulation for Au in the inset. The broad diffuse background ring originates from the amorphous ZrO$_2$ template.

**Table 3.** Average Au particle size (d) obtained from the TEM analysis and carbon content on the surface of the spent catalyst samples after the reaction.

| Catalyst | d (nm) | Carbon Content (wt.%) |
|---|---|---|
| 1 Au/ZrO$_2$ DP | 10.8 | 1.17 |
| 0.5 Au/ZrO$_2$ DP | 1.1 | 0.59 |
| 0.5 Au/ZrO$_2$ IMP | 50.1 | 0.92 |

### 2.2.7. XPS Analysis

Figure S4 (Supplementary Information) shows the survey spectra for the four powder samples. All of the survey spectra show signals for O 1s, Zr 3s, Zr 3p, and Zr 3d, corresponding to ZrO$_2$. A minor contribution to the O 1s signal also originates from the oxidized adventitious carbonaceous species. The C 1s signal, which was present in all of the survey spectra, originates from adventitious carbonaceous species adsorbed on the powder samples after the preparation procedure and during the transport of the sample to the XPS spectrometer. The C 1s signal disappeared when sputtering was performed using both 20 Ar$_{500}{}^+$ cluster and 5 keV Ar$^+$ sources, proving that the adventitious carbonaceous species were adsorbed on the surface of the powder samples. Samples that contained Au also show low intensity Au 4f peaks (for all of the survey spectra apart from the survey spectrum for ZrO$_2$).

The quantification of the relative atomic and weight concentrations was performed based on the measured high-resolution XPS spectra. The concentrations were normalized to 100.0%. The atomic ratio of Zr:O is close to 2:1, and therefore corresponds well to the ZrO$_2$ compound (Table S2, Supplementary Information). The slightly more than two-times

higher atomic concentration of O relative to Zr is due to the additional contribution of the oxidized carbonaceous species present on the surface (the high binding energy peak at 288.5 eV corresponding to the COO⁻/COOH species, and a high binding energy shoulder at 286.5 eV of the main peak in the C 1s spectra corresponding to the C-O/C=O species, Figure 8a). The positions of the Zr $3d_{3/2}$ and Zr $3d_{5/2}$ peaks are at 182.0 eV and 184.4, respectively, corresponding to a $ZrO_2$ compound. The shape and position of the peaks in the Zr 3d spectra of all four samples are at similar binding energies, indicating a similar environment of the Zr-containing compound, i.e., $ZrO_2$ (Figure 8b). The main peak in the O 1s spectra (Figure 8c) is located at 529.9 eV, and originates from $ZrO_2$. The high-energy shoulder of the main peak in the O 1s spectra originates from oxidized adventitious carbonaceous species (Figure 8c). Although the powder samples contained only a small amount of Au, the signal for Au 4f was detected for all of the samples (Figure 8d), with the highest Au content found in the 1 $Au/ZrO_2$ DP sample (Table S2). The relative atomic concentration of Au was the same for the 0.5 $Au/ZrO_2$ DP and 0.5 $Au/ZrO_2$ IMP samples, while this concentration was higher for the 1 $Au/ZrO_2$ DP solid (Table S2). Therefore, the atomic concentration ratio of Au:Zr was significantly higher for the sample of 1 $Au/ZrO_2$ DP than for samples of 0.5 $Au/ZrO_2$ DP and 0.5 $Au/ZrO_2$ IMP. Moreover, the weight surface concentration of Au reported in Table S2 for the 1 $Au/ZrO_2$ DP sample was higher than the actual loading. The deviation from the actual and measured Au atomic/weight percentages most likely means that Au is preferentially located on the surface of the powdered particles, and therefore more Au is detected in the XPS measurements (this technique obtains the signal from approximately 5 nm in depth). Concerning the samples of 0.5 $Au/ZrO_2$ DP and 0.5 $Au/ZrO_2$ IMP, no surface enrichment of the catalyst particles with gold ensembles was observed.

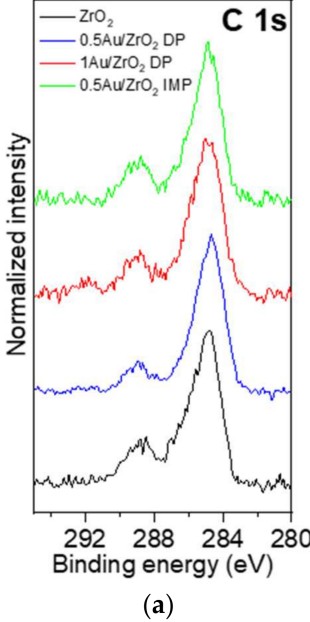
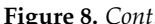
(a)

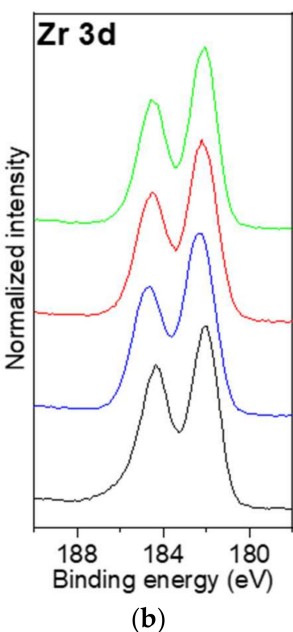
(b)

**Figure 8.** *Cont.*

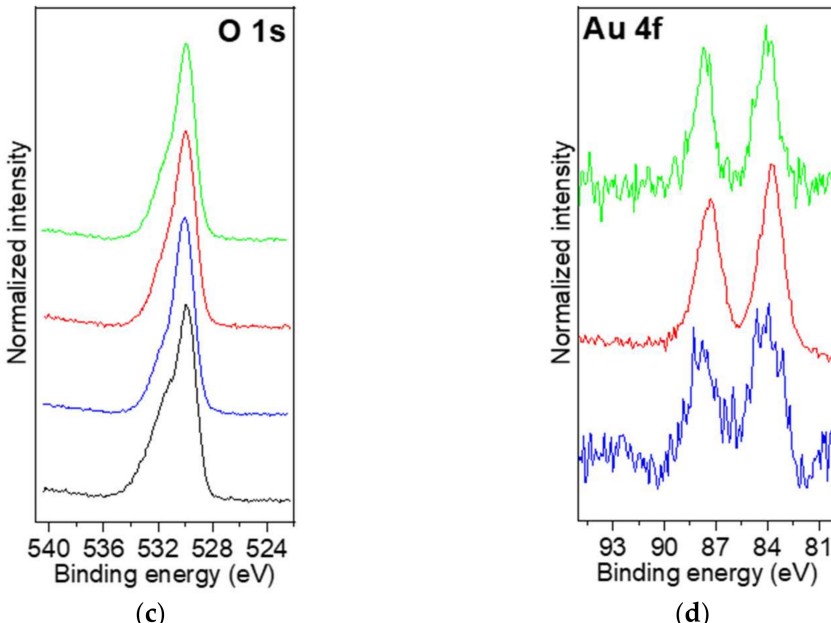

(c)  (d)

**Figure 8.** (**a**) C 1s, (**b**) Zr 3d, (**c**) O 1s, and (**d**) Au 4f high-resolution spectra for the $ZrO_2$ support and $Au/ZrO_2$ catalyst samples.

### 2.2.8. In Situ DRIFTS Analysis

In order to understand the formation of possible surface reaction intermediates, and to investigate the reaction mechanism of $CO_2$ hydrogenation process, in situ DRIFTS analysis at elevated pressures commenced. A steady-state experiment was performed in the presence of the 1 $Au/ZrO_2$ DP catalyst with the reaction mixture containing 72 vol.% $H_2$ and 24 vol.% $CO_2$ (balanced with $N_2$), which was purged on the pre-reduced catalyst for 40 min at 240 °C under 40 bar pressure. Later, the gas flow was changed to inert gas ($N_2$) flow for 80 min at the same temperature and pressure. The obtained results of the in situ DRIFTS analysis are presented in Figures 9–13 and S5–S9.

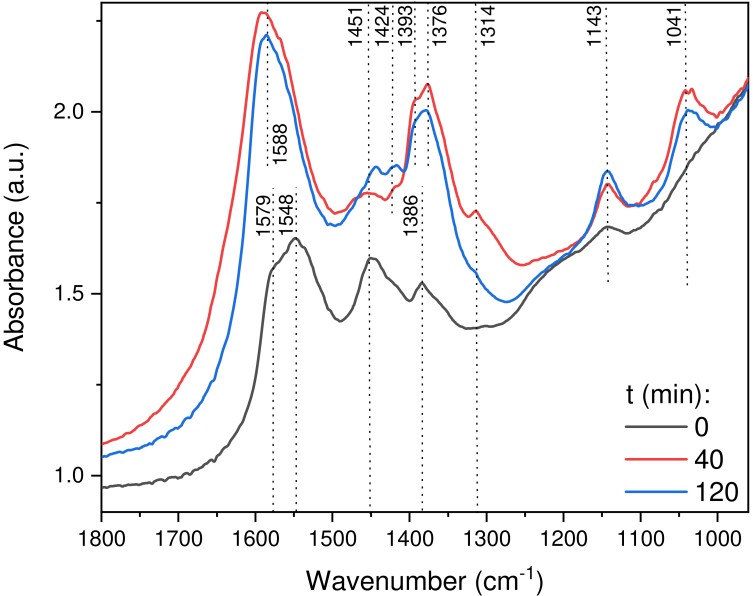

**Figure 9.** DRIFTS spectra showing the carbonate segment of the 1 $Au/ZrO_2$ DP catalyst under the reaction gas mixture and the $N_2$ gas switch after 40 min at T = 240 °C and $P_{tot.}$ = 40 bar.

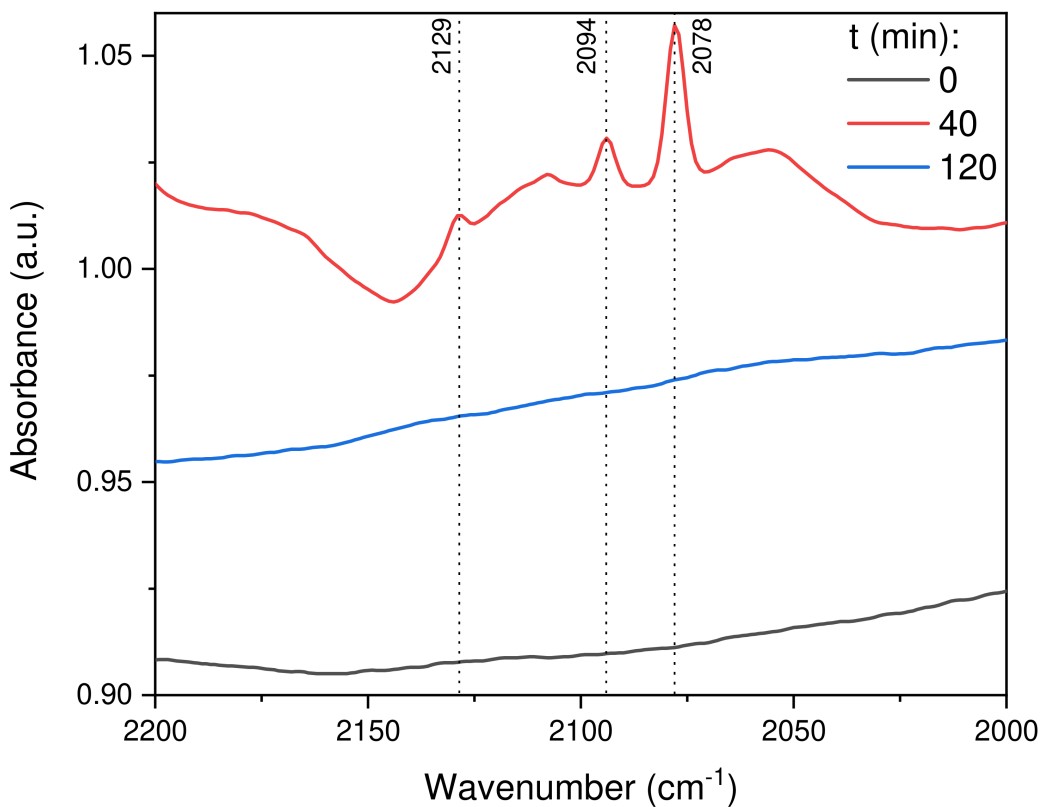

**Figure 10.** DRIFTS spectra showing the CO vibration segment of the 1 Au/ZrO$_2$ DP catalyst under the reaction mixture and the N$_2$ gas switch after 40 min at T = 240 °C and P$_{tot.}$ = 40 bar.

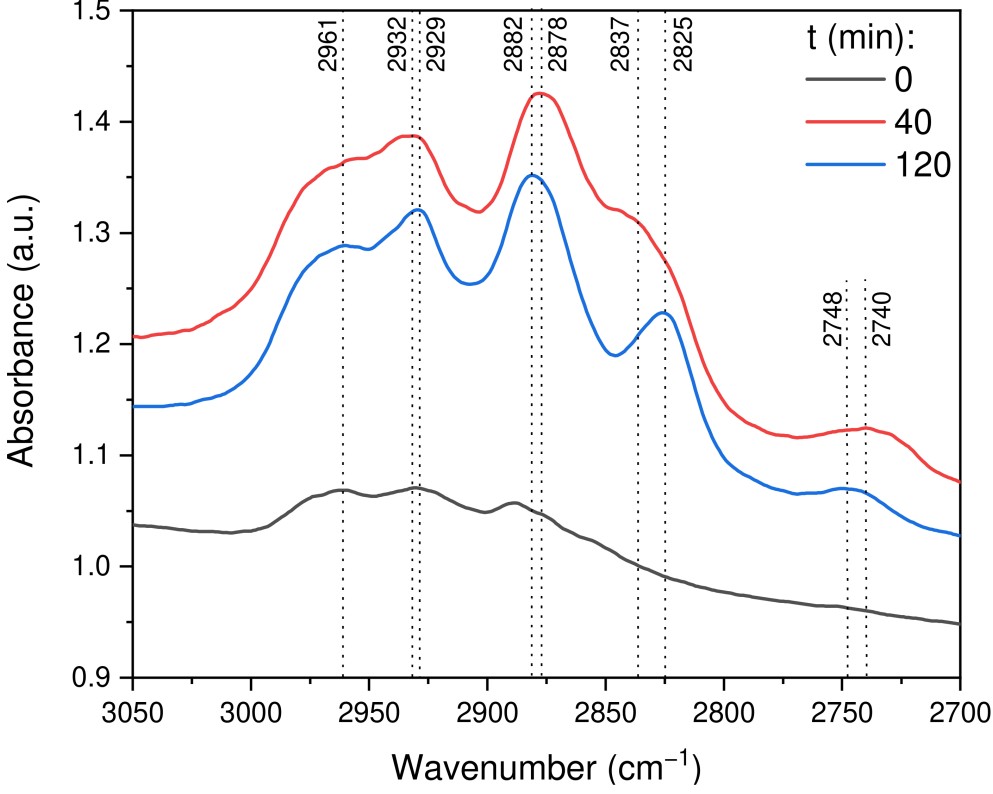

**Figure 11.** DRIFTS spectra showing the CH vibration segment of the 1 Au/ZrO$_2$ DP catalyst under the reaction gas mixture and the N$_2$ gas switch after 40 min at T = 240 °C and P$_{tot.}$ = 40 bar.

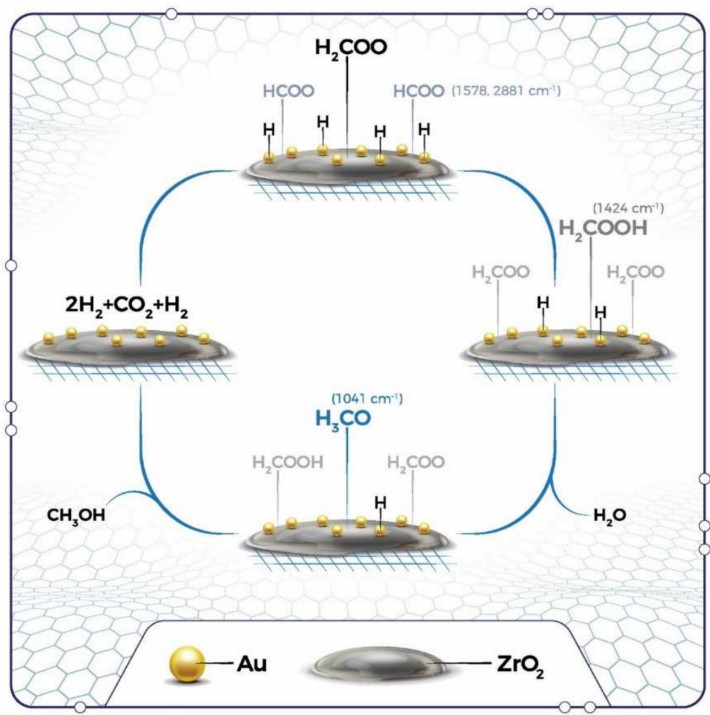

**Figure 12.** Proposed reaction mechanism of $CO_2$ hydrogenation to methanol carried out in the presence of Au-based $ZrO_2$ catalysts.

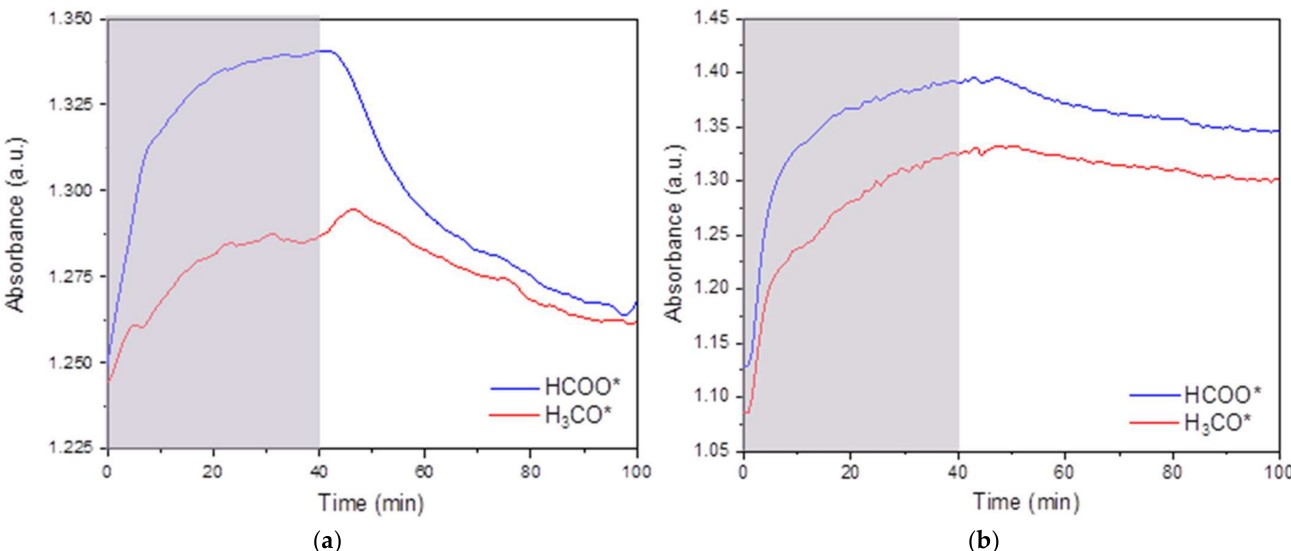

**Figure 13.** Temporal profiles for the HCOO* and $H_3CO$* surface intermediates obtained during the in situ DRIFTS measurements carried out in the presence of (**a**) 0.5 Au/$ZrO_2$ DP and (**b**) 0.5 Au/$ZrO_2$ IMP catalysts. The shaded part represents the time period of the $CO_2$/$H_2$ reaction mixture flow, and the white colour represents the time period of the $N_2$ flow.

The RMS intensity profile (the intensity value is the root-mean-square ordinate value in the spectrum drawn against time) of the entire experiment is illustrated in Figure S5 (Supplementary Information). The spectra recorded for the 1 Au/$ZrO_2$ DP catalyst under the reaction gas mixture from 0 to 21.75 min ("0 min" refers to the spectrum recorded before introducing the reaction mixture) at T = 240 °C and $P_{tot.}$ = 40 bar are shown in Figure S6 (Supplementary Information). Figure S6a shows the whole spectrum range from 4000 to 750 cm$^{-1}$, while Figure S6b–d indicates different important segments, namely the carbonate, CO and CH vibration segments, respectively. Figure S7 (Supplementary

Information) shows the spectra recorded for the 1 Au/ZrO$_2$ DP catalyst under an N$_2$ mixture from 60 to 120 min at T = 240 °C and P$_{tot.}$ = 40 bar. As mentioned previously, the whole spectrum from 4000–750 cm$^{-1}$ is presented in Figure S7a, and the important segments for the carbonate, CO and CH vibration modes are again split into Figure 7b–d, respectively. The numbers in Figures S5 and S6 indicate the time in minutes, and the arrows show changes in the intensities of the vibrational bands in the spectra with time. There were many spectra collected during the whole experiment; however, for a better understanding, we chose to present three important spectra: (i) at the beginning of the experiment (t = 0 min), before the introduction of the reaction mixture; (ii) exactly after the first gas switch from the reaction mixture to N$_2$ (at 40 min); and (iii) the final spectrum (collected at 120 min) under an inert gas (N$_2$) flow. FTIR measurements were performed in the spectral region of 4000–750 cm$^{-1}$, and the acquired spectra at 0, 40 and 120 min are presented in Figure S8 (Supplementary Information). In order to interpret in detail the full spectra, the latter were further divided into three meaningful segments.

A carbonate region from 1800 to 950 cm$^{-1}$ was considered as segment 1, which is presented in Figure 9. The initial spectrum at t = 0 min recorded before the introduction of the reaction mixture showed different carbonate bands appearing at 1579, 1451 and 1384 cm$^{-1}$, which were attributed to the OCO vibration bands of the symmetric and asymmetric stretching of bidentate carbonate species present in the formate intermediate (-OOCH) [40,42,60]. A vibrational band at 1548 cm$^{-1}$ further corresponded to the OC stretching vibrations of monodentate carbonate species present in the formate intermediate. This spectrum shows the initial formation of both bi- and monodentate formate intermediate species. A band at 1143 cm$^{-1}$ was also observed as a broad peak that appeared due to the stretching vibrations of OC in the OCH$_3$ intermediate species. After 40 min of the reaction mixture being fed, a considerable change was identified in the surface carbonates. The spectrum at 40 min shows high-intensity vibrational bands at 1588, 1448, 1394 and 1314 cm$^{-1}$ that were attributed to the formation of bidentate carbonate, whereas the formation of the monodentate OC stretching of OCH$_3$ was observed at 1143 cm$^{-1}$, and of the bidentate OC stretching of OCH$_3$ at 1041 cm$^{-1}$ [61]. Furthermore, a sharp shoulder peak occurred at 1378 cm$^{-1}$, which is commonly referred in the literature as the C-H vibrational band in the formed formate species. The gradual surface changes under the reaction mixture with time can be clearly observed in Figure S6b; these changes illustrate the formation of intermediates and the further hydrogenation of the formate species. After the reaction mixture treatment for 40 min, the gas flow was changed to N$_2$, and then a steady state was recorded at t = 120 min; the acquired spectrum shows a few more important intermediates. Along with different bidentate carbonate species, a new peak related to dioxymethylene (DOM) appeared at 1424 cm$^{-1}$ [62]. The formation of this intermediate species indicates the slow hydrogenation of the formed formate intermediate during the steady state under N$_2$ flow. Furthermore, the decrease of the 1041 cm$^{-1}$ peak related to methoxy species indicates the formation of methanol, and the increase of the 1143 cm$^{-1}$ peak suggests the formation of methoxy species due to the hydrogenation of surface formate intermediates. A steady growth in the vibrational bands of DOM and methoxy intermediates observed in Figure S7b illustrates the further hydrogenation of intermediates with time in the N$_2$ gas flow.

Segment 2, illustrated in Figure 10, represents the CO vibrational region. At t = 40 min, the catalyst showed vibrational bands at 2078, 2094 and 2129 cm$^{-1}$ that were assigned to surface-adsorbed CO on the Au surface [37,63,64]. No peaks were observed in this region at 0 and 120 min, i.e., at the beginning of the reaction and at the end of the experiment under an inert gas flow, because the formation of CO was mainly due to the side reaction (RWGS, Equation (2)). This particular side reaction was highly possible when the reaction mixture was flowing; hence, there were no peaks observed in this region at 0 and 120 min. However, the FTIR spectrum obtained at 40 min shows the above-mentioned peaks, indicating that this side reaction accompanied the CO$_2$ hydrogenation to methanol reaction even at 40 bar. Figure S6c shows that the CO vibrational bands progressively increase with time, which is attributed to the forward reaction in a CO$_2$/H$_2$ mixture flow. On the other hand, a

continuous decrease of CO vibrational bands can be observed in Figure S7c, indicating that CO is produced in the reaction conditions, such that one can conclude that CO is formed is mainly due to the RWGS reaction.

Furthermore, segment 3—displayed in Figure 11—shows peaks at 2961, 2932, 2878, 2837 and 2740 $cm^{-1}$ that were attributed to the absorption of IR radiation by the C-H vibrations of different intermediates. The initial spectrum acquired at 0 min also showed similar peaks but of lower intensities. The spectrum recorded at 40 min shows these peaks that were denoted in the previous literature reports as C-H vibrations in formate and methoxy intermediates. The combination of the peaks at 1578, 1443, 1380 and 1314 $cm^{-1}$ related to OCO vibrations, and the peaks occurring at 2881, 2974 and 2746 $cm^{-1}$ corresponding to CH vibrations, collectively represent the formation of the formate intermediate [40,61]. Similarly, the combination of the peaks at 1039 and 1142 $cm^{-1}$ appearing due to OC vibrations, and the peaks at 2931 and 2828 $cm^{-1}$ related to $CH_3$ vibrations, collectively indicate the formation of the methoxy intermediate [40,61]. In the spectrum acquired at 40 min, one can observe all of these peaks related to surface formate and methoxy intermediates. Figure S6d shows a gradual growth of CH vibrational bands, especially the band at 2828 $cm^{-1}$, which is attributed to the continuous hydrogenation of surface intermediates in order to form the methoxy intermediate. On the other hand, after the continuous flushing of the catalyst with $N_2$ for 80 min, the spectrum at 120 min showed an increased sharpness of the peaks at 2930 and 2825 $cm^{-1}$, which correspond to the methoxy intermediate, indicating that the hydrogenation of the surface formate species was continuously taking place even under the $N_2$ flow. Furthermore, continuous hydrogenation leads to the formation of a DOM intermediate, as observed in Figure 9; the same finding was supported with the occurrence of peaks at 1143 and 1041 $cm^{-1}$. A similar phenomenon can also be observed in Figure S7d, where vibrational bands related to CH vibrations continuously increase with time in an $N_2$ gas flow. This is another important observation that shows the hydrogenation of the intermediates even in an inert atmosphere.

Figure S9 (Supplementary Information) shows the temporal profiles of the important intermediates present on the catalyst surface. The increasing concentration of both the formate and methoxy intermediate species was observed under the flow of the reaction mixture ($H_2/CO_2$) through the DRIFTS cell. After the flow was changed to $N_2$, there was a continuous decrease observed in the surface concentration of formate and methoxy intermediates. However, the $H_3CO^*$ absorbance-time profile decreased slowly compared to the corresponding $HCOO^*$ profile. This observation suggests that the formation of both intermediates was indeed influenced by the reaction under consideration. Soon after the $H_2/CO_2$-to-$N_2$ gas switch, the continuous hydrogenation of formate species to the methoxy intermediate was noticed with the increase of the intensity of the methoxy intermediate vibrational bands (see Figure 11). Later, the continuous decrease of the absorbance as a function of time noticed in the same profiles suggested that the desorption of methoxy species from the catalyst surface occurred.

Based on the above observed surface intermediates and a thorough literature study, we propose a possible reaction mechanism (Figure 12) and a series of reaction steps (Equations (3)–(9)) for $CO_2$ hydrogenation to methanol via the formate route, and a series of reaction steps (Equations (S1)–(S7)) for the CO route (Supplementary Information). The formate route reaction steps are mainly towards the desired product, i.e., methanol, and no side reactions such as RWGS and CO hydrogenation are included. RWGS is an unwanted side reaction which is difficult to control, while CO hydrogenation requires high energies compared to $CO_2$ hydrogenation, according to the DFT calculations [40]. The adsorption of the reactants takes place on different surface sites, i.e., the adsorption of $CO_2$ on $ZrO_2$, and the adsorption of $H_2$ on Au ensembles and/or Au-Zr interfaces [65,66]. The dissociated H-H on Au and/or Au-Zr interfaces reacts with adsorbed $CO_2$, and results in the formation of formate species along with $H^*$. Furthermore, another $H_2$ molecule dissociates into $H^*$, which then participates in the hydrogenation of formate into $H_2COO^*$. As reported by Wang et al. [40], DFT calculations on ZnO-$ZrO_2$ show that the terminal protonation of

the $H_2COO^*$ intermediate is highly possible, and this reaction produces a highly reactive intermediate $H_2COOH^*$, which then cleaves at the C-O bond and generates $H_2CO^*$ and $OH^*$ species on the support and active metallic sites, respectively. After this, $OH^*$ reacts with $H^*$ and is eliminated from the catalyst surface as water. $H_2CO^*$ adsorbed on the $ZrO_2$ surface is further hydrogenated with the available $2H^*$ adsorbed on adjacent Au sites, and produces a stable intermediate: $CH_3O^*$. Finally, the hydrogenation of the methoxy intermediate into methanol takes place.

$$CO_2 + 2H_2 + H_2 \rightarrow HCOO^* + H^* + 2H_2 \tag{3}$$

$$HCOO^* + H^* + 2H_2 \rightarrow HCOO^* + 3H^* + H_2 \tag{4}$$

$$HCOO^* + 3H^* + H_2 \rightarrow H_2COO^* + 2H^* + H_2 \tag{5}$$

$$H_2COO^* + 2H^* + H_2 \rightarrow H_2COOH^* + H^* + H_2 \tag{6}$$

$$H_2COOH^* + H^* + H_2 \rightarrow H_2CO^* + 2H^* + H_2O \tag{7}$$

$$H_2CO^* + 2H^* \rightarrow H_3CO^* + H^* \tag{8}$$

$$H_3CO^* + H^* \rightarrow H_3COH \tag{9}$$

A similar experiment was conducted for both 0.5 wt.% $Au/ZrO_2$ catalysts synthesized by the DP and IMP methods. Important surface intermediates, i.e., formate and methoxy species, were observed in both cases. As such, we can conclude that a similar reaction mechanism is applicable for these catalysts as well. On the other hand, a close look at the temporal profiles for both the formate and methoxy intermediates (Figure 13) reveals that the profile belonging to the formate intermediate decreased rapidly in the case of the 0.5 $Au/ZrO_2$ DP catalyst, whereas the surface concentration of the formate intermediate decreased slowly in the case of the 0.5 $Au/ZrO_2$ IMP catalyst. This implies that the hydrogenation of formate species was very slow in the presence of the 0.5 $Au/ZrO_2$ IMP catalyst, and rapid in the case of the 0.5 $Au/ZrO_2$ DP catalyst. Similarly, the methoxy intermediate temporal profile shows an increase of absorbance after the reaction gas mixture's switch to $N_2$ in the case of the 0.5 $Au/ZrO_2$ DP catalyst compared to the 0.5 $Au/ZrO_2$ IMP catalyst. This can be explained by taking into consideration the presence of bulk Au clusters in the case of the 0.5 $Au/ZrO_2$ IMP catalyst, and of Au nanoparticles (~1 nm) in the case of the 0.5 $Au/ZrO_2$ DP sample, as observed by TEM analysis (Figure S3). As reported by Hartadi et al. [29], the surface richness of active metal species (Au) is highly dependent on the particle size of Au. As such, it is clear that the increase of the particle size decreases the surface richness of Au, which plays a vital role in the $CO_2$ hydrogenation to methanol reaction. Therefore, the small Au clusters present on the surface of catalysts prepared by the DP method improved the selectivity to methanol, whereas the bulk Au ensembles on the surface of the 0.5 $Au/ZrO_2$ IMP sample showed less efficiency for methanol production. According to Witoon et al. and Fisher and Bell, the $CO_2$ adsorption on the support and the hydrogenation to formate, DOM and methoxy species mainly depends on the strength of the basic sites [56,67]. As they mentioned, basic sites with low strength cause the rapid dissociation of formate to CO, whereas strong basic sites enable the further hydrogenation of the surface intermediates. Therefore, solids prepared by the DP preparation method with a high number of basic sites—especially of strong basic sites, i.e., $Zr^{4+}/O^{2-}$ centers ($\gamma$-values in Table 2)—play a vital role in entrapping $CO_2$ on the surface, and in further hydrogenation steps. On the other hand, the content of the $Zr^{4+}/O^{2-}$ centers highly decreases in the case of the catalyst prepared by the IMP method. The 1.5-times higher content of $Zr^{4+}/O^{2-}$ centers is observed on the surface of the corresponding solid prepared by the DP method rather than the IMP method, which enhances the adsorption of $CO_2$ as well as the further hydrogenation of the intermediates to methanol.

The CO formation is mainly due to the RWGS reaction; however, the decomposition of formate species can also produce CO and water.

Finally, the rate of methanol formation and the STY of methanol are reported in Figure 14; it can be observed that the values of both parameters are appreciably higher for the 0.5 $Au/ZrO_2$ DP catalyst compared to the 0.5 $Au/ZrO_2$ IMP solid, which is in agreement with the above discussion.

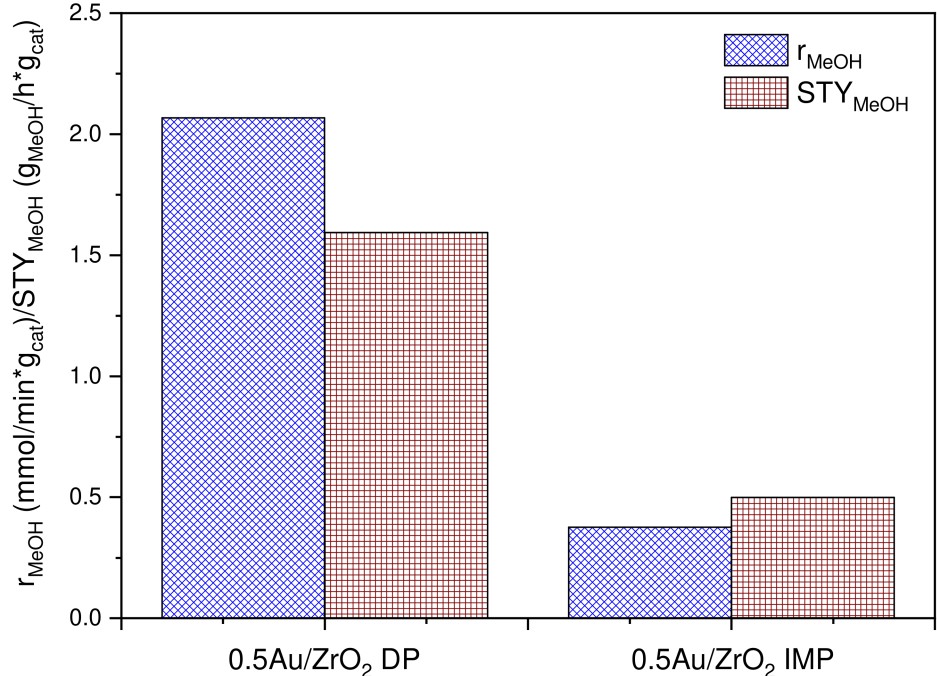

**Figure 14.** Rate of methanol formation and the STY of methanol for 0.5 wt.% $Au/ZrO_2$ catalysts synthesized by DP and IMP preparation procedures, and examined at T = 240 °C and $P_{tot.}$ = 40 bar.

2.2.9. CHNS Analysis of the Spent Catalysts

The amount of carbon which accumulated on the surface of spent catalyst samples after the reaction was determined by means of CHNS elemental analysis. These results showing nominal coke deposition on the catalyst are listed in Table 3. Rather large differences in coke formation were noticed for solids containing the same Au loading but synthesized by different catalyst preparation procedures. A higher coke resistance in the process of $CO_2$ hydrogenation to methanol is observed in the case of the 0.5 $Au/ZrO_2$ DP catalyst (0.59 wt.%) compared to the 0.5 $Au/ZrO_2$ IMP sample (0.92 wt.%). It was observed in the DRIFTS study that the rapid hydrogenation of intermediate species takes place in the presence of $Au/ZrO_2$ samples prepared by the DP method, which contributes to the removal of carbon-containing intermediates from the catalyst surface. On the other hand, the slow hydrogenation of formate and methoxy intermediates, as observed in the case of the 0.5 $Au/ZrO_2$ IMP catalyst, might be the reason for the noticeable coking of the catalyst surface. Analogously, it can be explained when comparing the 0.5 $Au/ZrO_2$ DP and 1.0 $Au/ZrO_2$ DP catalysts prepared by the same synthesis procedure, that the higher extent of coking in the latter case (Table 3) is due to the same reason (note that the reaction rate expressed per $g_{Au}$ is lower in the case of the 1.0 $Au/ZrO_2$ DP catalyst compared to the 0.5 $Au/ZrO_2$ DP solid). To conclude, carbon deposition can be minimized with the small particle size of the active metal; in this study, small Au particles were produced by the lowest loading of Au using the DP method. Furthermore, a large particle size with the increase in loading is observed due to the agglomeration of Au that resulted in the higher extent of coking.

### 3. Experimental Section

*3.1. Catalyst Synthesis*

3.1.1. Synthesis of the $ZrO_2$ Support

The $ZrO_2$ support was synthesized by the precipitation method using zirconium (IV) oxynitrate ($ZrO(NO_3)_2 \cdot xH_2O$, p.a., Sigma-Aldrich, Saint Louis, MO, USA) and aqueous ammonia as the precipitating agent. In total, 3 g zirconium (IV) oxynitrate was dissolved in ultrapure water and stirred on a magnetic stirrer for 30 min. Afterwards, the 5 vol.% ammonia was added dropwise until the complete precipitation of the Zr precursor. The precipitation of the zirconium salt was carried out at 25 °C and pH = 7. After complete precipitation, the support was aged in an oven for 24 h at 80 °C in air. Then, the solution was filtered and washed with ultra-pure water several times. The white precipitate was dried overnight at 90 °C. The obtained powder was further ground well and calcined at 300 °C for 4 h.

3.1.2. Au Deposition Precipitation on the $ZrO_2$ Support

The $Au/ZrO_2$ catalyst was synthesized by the deposition of 0.5 and 1 wt.% Au. In total, 1 mM gold (III) chloride trihydrate ($HAuCl_4 \cdot 3H_2O$, p.a., Sigma-Aldrich, Saint Louis (MO), USA) solution and 5% aqueous ammonia solution were used for the deposition precipitation method. In total, 1 g $ZrO_2$ support was mixed to the quantified amount of gold solution and then stirred for 30 min on the magnetic stirrer. The precipitation was carried out with the dropwise addition of aqueous ammonia solution at pH = 9. After the complete precipitation, the mixture was transferred into the oven and treated at 80 °C for 6 h. Later, the precipitated mixture was decanted, washed thoroughly, and dried for 2 h at 120 °C. Finally, the white solid was ground well and reduced at 350 °C for 2 h under a 5% $H_2$ balanced $N_2$ flow. The prepared catalyst samples were denoted as $xAu/ZrO_2$ DP (where x = 0.5 and 1.0 wt.%).

3.1.3. Au Impregnation on the $ZrO_2$ Support

The $Au/ZrO_2$ catalyst was synthesized by impregnation with 0.5 wt.% Au. In total, 1 mM gold (III) chloride trihydrate ($HAuCl_4 \cdot 3H_2O$, p.a., Aldrich, Saint Louis, MO, USA) solution was used for the catalyst preparation. In total, 1 g $ZrO_2$ support was mixed into the aqueous gold solution and then stirred for 2 h on a hot plate at room temperature. The hot plate's temperature was raised to 90 °C with a heating rate of 10 °C/min, and continuous heating was applied until the evaporation of excess water with stirring. The obtained solid was then dried overnight at 120 °C, and the powder was ground well and reduced at 350 °C for 2 h under a 5% $H_2$ balanced $N_2$ flow. The prepared catalyst was denoted as 0.5 $Au/ZrO_2$ IMP.

*3.2. Catalyst Characterization*

The specific surface area, total pore volume and average pore diameter of the reduced catalysts were determined using $N_2$ adsorption–desorption isotherms. These analyses were carried out on a TriStar II 3020 instrument (Micromeritics, Norcross, GA, USA). Before the measurements, the degassing of the samples was carried out by a Smart-Prep degasser (Micromeritics, Norcross, GA, USA) in $N_2$ gas at 90 °C for 1 h, and then at 120 °C for 4 h. The total pore volume and pore size were calculated by the Barrett-Joyner-Halenda (BJH) method from the desorption isotherms.

Inductively coupled plasma optical emission spectroscopy (ICP-OES) analysis was conducted for the reduced catalyst samples in order to determine the content of Au by using a Varian 715-ES ICP optical emission spectrometer (Santa Clara, CA, USA).

The powder X-ray diffraction (XRD) measurements of the samples (reduced catalysts) were carried out on a PANalytical X'Pert PRO MPD diffractometer (Almero, The Netherlands) with Cu Kα1 radiation (λ = 1.5406 Å) in the 2θ range from 5 to 90°, with the 0.034° step every 1000 s using a fully opened X'Celerator detector. Xpert high score plus version 2.2.3 software was used to interpret the experimental diffractograms. The temperature-

programmed in situ reduction of samples was carried out on a PANalytical X'Pert PRO diffractometer (Almero, The Netherlands) with Cu K$\alpha$ radiation ($\lambda$ = 1.541874 Å) in the temperature range from 25 to 500 °C with a stepwise temperature increment of 25 °C and a heating ramp of 10 °C/min under 5% $H_2$ balanced Ar flow. The diffractograms were recorded in the 2$\theta$ range from 10 to 80°, with the 0.039° step every 100 s using a fully opened PIXCel detector.

The morphology, size and crystal structure of the nanomaterials was analyzed by transmission electron microscopy (TEM, JEM-2100, Jeol, Tokyo, Japan), operating at 200 kV and equipped with a high-resolution slow-scan CCD camera (Orius SC 1000, Gatan, Pleasanton (CA), USA). The powdered samples were dispersed in absolute ethanol and sonicated to prevent agglomeration, and the suspension was transferred onto Cu-supported amorphous carbon grids. The Au nanoparticles' size was determined from their Feret size [34], measured directly from the TEM micrographs. Before the TEM investigation, the microscope image and diffraction mode were calibrated by the MAG*I*CAL® reference standard through all of the magnification ranges, with the overall uncertainty on the calibrated values $\Delta t$ < 1.0%.

A Supra+ XPS instrument (Kratos, Manchester, UK) was used for the XPS measurements. An Al K$_\alpha$ and a monochromator were used as the excitation source. The measurements were performed on a 300 × 700 micron spot at a 90° take-off angle. The survey and high-resolution spectra were measured at a pass energy of 160 and 20 eV, respectively, in the hemispherical analyzer. The charge neutralizer was on during the measurements. The emission current for the survey and high-resolution spectra measurements was 15.00 mA. The data were acquired and processed using ESCApe (Kratos, Manchester, UK). The binding energy scale was corrected using the C-C/C-H peak in the C 1s spectrum, located at 284.8 eV. Sputtering was performed using monoatomic 5 keV $Ar^+$ and 20 $Ar_{500}^+$ clusters (the clusters were produced using the gas cluster ion source ion gun).

The nature of basic sites and total basicity was analyzed by the $CO_2$-TPD technique using the Micromeritics AutoChem II 2920 apparatus (Norcross, GA, USA) connected to a mass spectrometer (Pfeiffer Vacuum, Aßlar, Germany, model ThermoStar$^{TM}$ GSD320). In total, ~55 mg of a reduced catalyst was used for the $CO_2$-TPD analysis. Although the catalysts were reduced before the analysis, the catalyst sample was pretreated in 5% $H_2$-balanced Ar at 350 °C for 30 min in order to produce a fresh reduction surface. The sample was then cooled to 50 °C, followed by pulses with 80% $CO_2$ (balanced with Ar) until the complete saturation of the surface. Then, the sample was purged for 30 min with He to remove physisorbed $CO_2$. The furnace temperature was raised to 350 °C with a heating rate of 5 °C/min, and $CO_2$ desorption was monitored by the mass spectrometer following the characteristic $m/z$ = 44 fragment. The obtained curve was deconvoluted using PeakFit software (Version 4.11).

Pyridine TPD experiments were conducted in order to understand the total acidity and nature of acidic sites on the surface of reduced catalysts by using a TGA instrument (Perkin Elmer, Waltham, MA, USA, model Pyris 1). A Pt pan was tared to 0 mg, then ~20 mg of the reduced catalyst was loaded into the pan attached to the TGA device. Before the analysis, the samples were treated in a $N_2$ flow at 120 °C for 30 min. The pyridine-saturated $N_2$ stream was then purged onto the sample until surface saturation at 120 °C. After that, the sample was treated with $N_2$ flow for over 1 h to remove the physisorbed pyridine. Then, the temperature was raised to 700 °C and kept at the final temperature for 5 min. The whole experiment was monitored according to the change in weight; later, the first derivative of weight was considered and presented as a Py–TPD curve.

UV-Vis diffuse reflectance spectra were recorded using a Lambda 650 instrument (Perkin Elmer, Waltham, MA, USA) equipped with a Praying Mantis reaction chamber (HVC-VUV) supplied by Harrick Scientific Products (Pleasantville, NY, USA). These measurements were conducted in the wavelength range between 200 and 800 nm. The background correction was carried out with a white reflectance standard provided by Spectralon©.

Concerning DRIFTS analysis, 50 mg of a catalyst was loaded in a porous ceramic alumina cup and placed in the DiffusIR cell (PIKE Technologies, Fitchburg, WI, USA) attached to Fourier-transform infrared spectroscopy (FTIR) device (Perkin Elmer, Waltham, MA, USA, model Frontier). Before the in situ analysis, the background was recorded with the metal plate supplied with the instrument. Then, the catalyst sample was pretreated with $H_2/N_2$ at 350 °C for 30 min. After that, the reaction cell was purged with $N_2$ gas, the temperature was decreased to 240 °C, and the pressure was elevated to 40 bar. DRIFTS experiments were carried out at 240 °C and 40 bar with the flow of reaction gas mixture, i.e., 24 vol.% $CO_2$ and 72 vol.% $H_2$-balanced $N_2$ for 40 min. After reaching a steady state, the gas switched to a $N_2$ flow at the same temperature and pressure.

Spent catalyst samples were analyzed using a Perkin Elmer CHNS analyzer (Waltham, MA, USA, model 2400 Series II) to determine the extent of the coking. Prior to the analysis, the samples were treated with 1.0 M HCl to remove inorganic carbon-containing compounds (i.e., carbonates and bicarbonates).

### 3.3. Activity Tests

The $CO_2$ hydrogenation to methanol activity tests were conducted in a high-temperature, high-pressure, computer-controlled reactor system (PID Eng&Tech, Madrid, Spain) equipped with a stainless-steel tubular fixed-bed reactor supplied by Autoclave Engineers. In total, 50 mg of a catalyst homogeneously mixed with 200 mg SiC was loaded at the center of the reactor tube with the help of an inserted stainless-steel frit (2 μm pores), onto which 60–70 mg of quartz wool was placed. The catalytic bed volume was equal to 0.2 mL. The catalyst was first pretreated at 350 °C and atmospheric pressure for 30 min in a 15% $H_2/N_2$ flow, and then the reactor temperature was decreased to 240 °C and the pressure increased to 40 bar in a $N_2$ flow. Afterwards, the gas reaction mixture containing 24 vol.% $CO_2$ and 72 vol.% $H_2$ (balanced with $N_2$) was introduced into the reactor with the total feed flowrate of 100 mL/min. The outlet gas stream was analyzed online using a gas chromatograph (Agilent Technologies, Santa Clara (CA), USA, model 7890A,) equipped with Porapak Q, HayeSep Q and MS5A columns. The $CO_2$ conversion, methanol selectivity, and space–time yield (STY) of methanol and the rate of methanol production were calculated by the following equations:

$$X_{CO_2} = \frac{CO_{2in} - CO_{2out}}{CO_{2in}} \times 100 \tag{10}$$

$$S_{CH_3OH} = \frac{CH_3OH_{out}}{CH_3OH_{out} + CO_{out}} \times 100 \tag{11}$$

$$STY_{CH_3OH} = \frac{CO_{2in} \times X_{CO_2} \times S_{CH_3OH} \times MW}{W_{Cat}} \tag{12}$$

$$r_{CH_3OH} = \frac{CH_3OH_{out}}{W_{cat}} \tag{13}$$

where $CO_{2in/out}$, $CH_3OH_{out}$ and $CO_{out}$ stand for the concentration of these compounds in mol/L, MW is the molecular weight of methanol, and $W_{cat}$ is the catalyst loading used for the reaction.

## 4. Conclusions

In conclusion, $Au/ZrO_2$ catalysts synthesized by the DP method exhibited less pore blockage compared to the catalyst synthesized by the IMP method, which in turn helped to retain their specific surface area. This pore blockage was mainly due to the formation of bulk clusters of Au ensembles in the 0.5 $Au/ZrO_2$ IMP sample, whereas the 0.5 $Au/ZrO_2$ DP catalyst delivered Au nanoparticles of ~1 nm. The enrichment of the catalyst surface with Au ensembles was observed by XPS analysis in the case of higher metal loadings. The enhanced metal–support synergy could be identified by several factors, i.e., (i) the amorphous-to-cubic phase transformation of $ZrO_2$, (ii) the enhanced strength of the acidic sites, and (iii) the increased number of basic sites. The addition of Au induced a phase

transformation from the amorphous to the cubic phase of $ZrO_2$ in both of the utilized catalyst preparation methods; however, the 0.5 Au/$ZrO_2$ DP catalyst exhibited a higher impact on the phase transformation. The strength of the acidic sites was increased with an increase of the Au content. The increased number of basic sites provoked by the Au deposition indicates an enhanced synergy between the metal and the support. However, compared to the catalyst synthesized by the IMP method, DP-based catalysts showed a higher number of basic sites. In situ DRIFTS experiments showed the different intermediate species present on the catalyst surface; the most important of them were HCOO*, DOM and $H_3CO^*$. By considering these intermediates, a reaction mechanism of $CO_2$ hydrogenation to methanol was developed via the formate route. Here, the time-resolved FTIR profiles of both 0.5 wt.% Au-loaded catalysts showed large differences in the hydrogenation of the formate and methoxy intermediates with respect to the catalyst preparation method. The high $CO_2$ uptake and the presence of small Au nanoparticles enabled the rapid hydrogenation of intermediate species to methanol over DP-based catalysts, whereas the low $CO_2$ uptake and the presence of bulk Au clusters on the catalyst surface lead to the slow hydrogenation of intermediate species to products, as observed in the case of the IMP-based catalyst. This slow hydrogenation, along with the increased size of the Au ensembles, may be the reason for the enhanced coke formation. Therefore, Au-supported $ZrO_2$ catalysts synthesized by the DP method were advantageous in the process of $CO_2$ hydrogenation to methanol, compared to the IMP-based catalyst.

**Supplementary Materials:** The following supporting information can be downloaded at: https://www.mdpi.com/article/10.3390/catal12020218/s1, Table S1: Comparison of performance of Au/$ZrO_2$ solids examined in this study with the reported Au supported $ZrO_2$ and modified $ZrO_2$ catalysts for $CO_2$ hydrogenation to methanol; Table S2: Relative atomic and relative weight surface concentration for the $ZrO_2$ support and Au/$ZrO_2$ catalysts calculated based on the measured high-resolution XPS spectra; Figure S1: (**a**) Catalytic activity of 0.5 Au/$ZrO_2$ DP catalyst obtained at different temperatures, and (**b**) results of the stability test carried out at 240 °C. Operating conditions: $P_{tot.}$ = 50 bar, catalyst weight: 50 mg (homogeneously mixed with 200 mg of SiC), gas flow rate: 100 ml/min, composition of the feed gas stream: 24 vol.% $CO_2$, 72 vol.% $H_2$ (balanced with $N_2$); Figure S2: UV-Vis DR spectra of $ZrO_2$ support and Au/$ZrO_2$ deposition precipitation/impregnation catalysts after reduction; Figure S3: TEM images of Au/$ZrO_2$ deposition precipitation/impregnation catalysts after reduction; Figure S4: XPS survey spectra measured for the $ZrO_2$ support and Au/$ZrO_2$ catalyst samples; Figure S5: RMS profile of time resolved DRIFTS spectra for the whole steady-state experiment region conducted over 1 Au/$ZrO_2$ DP catalyst under reaction mixture and $N_2$ gas switch after 40 min at 240 °C and 40 bar total pressure; 1. Reaction steps of $CO_2$ to methanol hydrogenation via CO route; Figure S6: DRIFTS spectra recorded for 1 Au/$ZrO_2$ DP catalyst under reaction gas mixture at T = 240 °C and $P_{tot.}$=40 bar: (**a**) the overall spectra, (**b**) the carbonate segment, (**c**) CO vibration segment and (**d**) CH vibration segment. The numbers 0 to 21.75 indicate time in minutes; Figure S7: DRIFTS spectra recorded for 1 Au/$ZrO_2$ DP catalyst under $N_2$ mixture at T = 240 °C and $P_{tot.}$ = 40 bar: (**a**) the overall spectra, (**b**) the carbonate segment, (**c**) CO vibration segment and (**d**) CH vibration segment. The numbers 60 to 120 indicate time in minutes; Figure S8: Selected DRIFTS spectra acquired at 0, 40 and 120 min by either using reaction gas mixture or $N_2$ gas, respectively, in the presence of 1 Au/$ZrO_2$ DP catalyst. Operating conditions: T = 240 °C, $P_{tot.}$ = 40 bar; Figure S9: Temporal profiles for HCOO* and H3CO* surface intermediates obtained during the in-situ DRIFTS measurements carried out in the presence of 1 Au/$ZrO_2$ DP catalyst; References.

**Author Contributions:** T.V.S.: Conceptualization, Investigation, Methodology, Writing—original draft, Writing—review and editing; J.Z.: Investigation, Writing—review and editing; M.F.: Investigation, Resources, Writing—review and editing; N.N.T.: Supervision, Writing—review and editing; A.P.: Conceptualization, Funding acquisition, Methodology, Project administration, Supervision, Writing—review and editing. All authors have read and agreed to the published version of the manuscript.

**Funding:** The authors acknowledge the financial support from the Slovenian Research Agency (project No. J7-9401 and Research Core Funding No. P2-0118). The project is co-financed by the Republic of Slovenia, the Ministry of Education, Science and Sport, and the European Union under the European Regional Development Fund.

**Conflicts of Interest:** The authors declare that they have no known competing financial interests or personal relationships that could have appeared to influence the work reported in this paper.

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
