# Peer review of "Evaluation of Au/ZrO2 Catalysts Prepared via Postsynthesis Methods in CO2 Hydrogenation to Methanol"

_catalysts, doi:10.3390/catal12020218_

Round 1
Reviewer 1 Report
The manuscript reports Au supported over zirconia catalyst for CO2 hydrogenation for methanol production. The synthesized catalyst is characterized well for morphology, surface area, composition, crystal structure, active metal sites and acidic-basic characteristics of the samples. The reaction was well characterized for the mechanism using insitu FTIR. The data acquired by in-situ technique support the reaction mechanism proposed. Overall the manuscript describes some interesting findings with appropriate discussions based on characterization data, however, the following comments for discussion and clarification should be considered before accepted for publication.
Comments:
- Section 3.2.3 and 3.2.4; CO2-TPD and Pyridine-TPD: What should appropriate temperature range for considering acidic or basic sites as weak, medium and strong sites? For acidic sites, temperature more than 600 °C is considered for strong sites but for basic sites, 200 °C is the temperature taken to indicate strong basic sites.
- Lewis and Brønsted character of the acidic and basic sites was not considered in adsorption of reactants and formation of intermediates. As both Lewis acid/base and Brønsted acid/base sites will not lead to formation of similar intermediate.
- CO2 is desorbed from the basic sites up to ~200 °C (third peak in Fig 6). Is this the reason why very low CO2 conversions were observed at operating temperature (240 °C)? Because CO2 is not adsorbed on the catalyst surface at this temperature.
- Insitu FTIR: It has clearly shown formation of CO over Gold nanoparticles. Additionally proposed mechanism in Fig 12 shows dissociative adsorption of hydrogen over Au nanoparticles and hydrogen spill over to support sites for hydrogenation of adsorbed CO2 intermediates. Hence, CO may lead to poising of gold particles and interfere in hydrogen adsorption leading very low conversion of CO2 as observed in the manuscript. Please see if this is appropriate.
Reviewer 2 Report
The manuscript describes the method to “Evaluation of Au/ZrO2 catalysts prepared via post-synthesis methods in CO2 hydrogenation to methanol”. The structural, morphological, and e hydrogen evolution properties of the samples were studied and compared. After careful evaluation of the paper, I recommend publication subject to a minor revision in the following aspects.
XRD results should be revised and compared with previous results of structural properties.
The authors should provide the FT-IR results and compare them to other properties.
Please use the same format for unit and symbols
Please polish the language carefully.
Standard JCPDS cards can be included the in XRD section
Annotations should be uniform for all figures. e.g. Fig.3 and Figures
The scale bar for TEM is unclear, size can be increased.
The alignment of figures is inappropriate. All figures should be aligned properly.
Morphology and structure analysis should be revised.
